# *Look-back* Decoding for Open-Ended Text Generation

**Nan Xu$^{\diamond}$, Chunting Zhou$^{\spadesuit}$, Asli Celikyilmaz$^{\spadesuit}$, Xuezhe Ma$^{\diamond}$**
$^{\diamond}$University of Southern California, $^{\spadesuit}$Meta AI
$^{\diamond}${nanx,xuezhema}@usc.edu, $^{\spadesuit}${chuntinz,aslic}@meta.com

## Abstract

Given a prefix (context), open-ended generation aims to decode texts that are coherent, which do not abruptly drift from previous topics, and informative, which do not suffer from undesired repetitions. In this paper, we propose *Look-back*, an improved decoding algorithm that leverages the Kullback–Leibler divergence to track the distribution distance between current and historical decoding steps. Thus *Look-back* can automatically predict potential repetitive phrase and topic drift, and remove tokens that may cause the failure modes, restricting the next token probability distribution within a plausible distance to the history. We perform decoding experiments on document continuation and story generation, and demonstrate that *Look-back* is able to generate more fluent and coherent text, outperforming other strong decoding methods significantly in both automatic and human evaluations[1].

## 1 Introduction

Despite the impressive success on generating fluent and accurate sentences for low-entropy tasks such as summarization or translation, large-scale language models (LLMs) still suffer from serious degeneration problems, such as undesired repetitions (Holtzman et al., 2019) and unnatural topic drifts, under open-ended settings (Eikema and Aziz, 2020). Open-ended neural text generation aims to generate coherent and diverse text from LLMs, given contextual prefix (Nadeem et al., 2020; Dhamala et al., 2022), and has spawned a wide range of natural language applications, including contextual text completion (Radford et al., 2019), story generation (Fan et al., 2018), and review generation (Cho et al., 2019).

To alleviate the degeneration problem in open-ended text generation, a number of techniques have

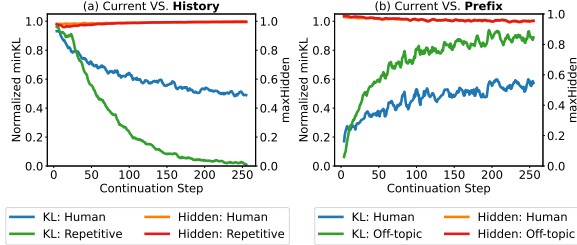

Figure 1: Maximum similarity of hidden states and normalized minimum KL divergence between current step and **history** (a) or **prefix** (b) from GPT2 on 1,000 instances of WikiText-103. Compared with human continuation, (a): *repetition* has much smaller *minKL* but undistinguishable high *maxHidden* with history text, (b): pseudo *topic drift* by switching to continuation of another instance has much higher *minKL* but similar high *maxHidden* with prefix text.

emerged over the recent years, which can be categorized into two directions: i) *improved learning* proposing new learning objectives, e.g., unlikelihood training (Welleck et al., 2019), contrastive training (Su et al., 2022) and sequence likelihood calibration (Zhao et al., 2022), to compensate for the rooted deficiency of the conventional Maximum Likelihood Estimation (MLE) [2]; ii) *improved decoding* remedying tedious and repetitive generations in decoding search (Su et al., 2022; Li et al., 2022), or combating topic drifts in sampling procedures (Hewitt et al., 2022).

In this work, we propose a new decoding algorithm, named *Look-back*, which pays particular attention to the probability distribution disparity between continuation and history text. Unlike contrastive search (Su et al., 2022; Su and Xu, 2022) which uses cosine similarity between the hidden representation, *Look-back* leverages the Kullback-Leibler (KL) divergence to track the distribution distance between current and historical decoding steps. The main motivation of *Look-back* is that

---

[1]Code and resources are available at https://github.com/xunannancy/LookBackDecoding.

[2]The correlation between sequence probability and its quality for MLE trained models can be low (Liu et al., 2022).

KL divergence defines a distance between the probability distributions of decoding steps, which arguably better aligns with the decoding practice. As shown in Figure 1 (a), as the greedy algorithm repeatedly outputs single sentences, the distance with the closest past token distribution decreases towards 0. Besides, when the continuation switches to another topic in Figure 1 (b), the distribution distance of continuation with prefix obtains much higher levels compared with topic-relevant human continuation. Based on our prior observations, for informative and coherent generation, the probability distribution should not be too close to history to guarantee diversity, but relatively close to prefix to maintain coherence.

Experimentally, through two tasks of open-ended text generation, including document continuation and story generation, we demonstrate that *Look-back* outperforms a variety of open-ended decoding algorithms under different scales of pre-trained LLMs (GPT2-XL and OPT-6.7B) by producing much more coherent texts – high mauve score compared with human continuation and high similarity score measured against prefix, while maintaining similar level of diversity.

## 2   Related Work

**Improved Learning Algorithms** Yang et al. (2018); Adiwardana et al. (2020) observed that increasing number of candidates in beam search or sampling leads to worse quality of generated data. They attribute this to the predominant training objective (i.e., Maximum Likelihood Estimation) that might not accurately rank generated sequences by quality (Zhao et al., 2022). Besides, Holtzman et al. (2019) found that searching for the probable sequences always results in short and repetitive texts, which further motivated recent efforts to improve generation via revised learning objectives. Welleck et al. (2019) proposed unlikelihood training to force unlikely generations to be assigned lower probability by the model. To alleviate degeneration, SimCTG (Su et al., 2022) introduced a contrastive training objective to preserve sparseness of the token similarity matrix of the generated text. To avoid unintentionally boosting the probability of other irrelevant tokens in unlikelihood training, Jiang et al. (2022) leveraged contrastive token learning to explicitly teach the LLM to assign negative tokens with a lower probability than positive tokens through more focused contrast between the two. Based on a BERTScore-style similarity metric between model decodes and targets measured in the model's latent space, Zhao et al. (2022) calibrated model-generated sequences with sequence likelihood calibration to better align with reference sequences via different types of losses (e.g., rank and margin loss).

**Improved Decoding Algorithms** Liu et al. (2022) observed that search methods (e.g., greedy and beam) which optimize generation probabilities may result in tedious and repetitive outputs in open-ended text generation. Su et al. (2022) complemented the contrastive training with contrastive search for decoding, which selects tokens more distinguishable from previous context. Li et al. (2022) observed that degeneration is more prevalent in larger LMs than smaller ones, and proposed contrastive decoding to remove these undesired behavior by factoring out smaller LM's behavior from the larger LM. On the other hand, truncation sampling methods such as nucleus (Holtzman et al., 2019) and typical (Meister et al., 2022) decoding improve sample quality with more diverse samples compared to direct sampling, but at the expense of poor coherence and undesired topic drift. Hewitt et al. (2022) introduced $\eta$-sampling to truncate words below an entropy-dependent probability threshold. A concurrent work observed the strong correlation between good generation quality and narrow entropy zone, hence proposed entropy-aware decoding to promote good generation by constraining greedy decoding into the narrow entropy zone (Arora et al., 2023).

Without extra effort on fine-tuning LMs, the proposed *Look-back* improves conventional search method with reference from the given prefix and prior generation, so that undesired repetitions and topic drifts can be explicitly alleviated.

## 3   Background

### 3.1   Open-ended Text Generation

Given a sequence of $m$ tokens sampled from natural text $\mathcal{C} = \{x_1 \ldots x_m\}$ as context or prefix, the neural text generation is to decode a $n$-token continuation using the probability distribution provided by pre-trained LMs:

$$p(x_{m+1:m+n}|\mathcal{C}) = \prod_{t=1}^{n} P(x_t|\mathcal{C}, x_{m+1} \ldots x_{m+t-1}),$$

| Degeneration | LM (Decoding) Continuation |
|---|---|

**Prefix**: In addition to live broadcasts FIFA Fan Fests offer food and beverages, merchandise and various entertainment events by local and international artists. The start of 2006 World Cup was

| | |
|---|---|
| **Repetitive** Continuation | *S1*: **(GPT2-XL greedy)** marked by the FIFA Fan Fest in Sao Paulo, Brazil.\n\n *The FIFA Fan Fest is a unique event that brings together fans from all over the world to celebrate the FIFA World Cup* . *The FIFA Fan Fest is a unique event that brings together fans from...* |
| | *S2*: **(davinci-002 greedy)** celebrated with a concert by Bon Jovi in Berlin.\n\n *The FIFA Fan Fest in* Munich *was located at the* Olympic Park.\n\n *The FIFA Fan Fest in* Frankfurt *was located at the* Römerberg.\n\n *The FIFA Fan Fest in* Hamburg *was located at the* ... |

**Prefix**: Burkan died from an attack of acute indigestion at his country home in Great Neck on June 6, 1936. Three thousand people attended his funeral at

| | |
|---|---|
| **Off-Topic** Continuation | *S3*: **(ada-001 greedy)** the local church.\n\n *The authorities are investigating how Khedira managed to enter the house and what role he played in the attack.* |
| | *S4*: **(davinci-002 greedy)**: Temple Emanu-El in New York City... *Category:1868 births\nCategory:1936 deaths\nCategory:Austro-...* |
| | *S5*: **(ada-001 nucleus)**: aients home.\n ( *The Lorraine weekend\nIn house of intensity and occupation, great law enforcement officers\n* ... *Shanny Bankecived his way into the home of Runaan U Without giving any reason other than to marines and punch said home's door* ... ) ×**2** |

Table 1: Degeneration examples with typical decoding algorithms by GPT2-XL and GPT3 (ada-001 and davinci-002). Complete sentence repetition (*S1*), repetition with minor location changes (*S2*) or paragraph duplication (*S5*) is marked in *green* , while unnatural (*S3*&*S4*) or stiff (*S5*) topic drifts are in *pink* .

where the continuation is generated token-by-token using a particular decoding strategy. For instance, greedy algorithm selects the next token given context with the highest probability, while nucleus sampling (Holtzman et al., 2019) restricts the plausible area of tokens with total mass above a threshold.

## 3.2 Degeneration Problems

There are two commonly observed degeneration problems in open-ended text generation: repetition and incoherence.

**Repetition** LLMs prefer to overestimate the probability of repeated sequences (Welleck et al., 2019) especially for deterministic algorithms such as greedy and beam search. Although decoding algorithms such as nucleus sampling (Holtzman et al., 2019) have been proposed to interrupt repeating sequences, we can still observe repetitive and tedious continuation even from the state-of-the-art GPT-3 language model (Brown et al., 2020), as shown in Table 1. Besides the consensus that probabilities from conditional LMs often do not accurately rank-order generated sequences by quality (Zhao et al., 2022), a recent study provides a possible way to explain the repetitive generation with the observed analogical sequence copying pattern: prefix matching and copying[3] (Olsson et al., 2022).

**Incoherence** Sampling algorithms sacrifice coherence for alleviating repetition during decoding.

---

[3]Prefix matching: the attention mechanism in transformer-based LMs attends back to previous tokens that were followed by the current and/or recent tokens. Copying: outputs increased logit of the attended-to token or others similar in embedding space.

As shown in Table 1, given probabilities from GPT-3 models, nucleus sampling fails to produce coherent generation, switching topic from *Burkan's acute indigestion* to *Shanny's way to home* with *ada-001* (*S5*). Recent decoding algorithms depend on model confidence to "guarantee" coherence while resolving repetition explicitly with certain heuristics. For example, SimCTG (Su et al., 2022) selects from most probable candidates predicted by LM. Contrastive decoding (Li et al., 2022) exploits coherence nature of the expert LMs. In both *S3* and *S4* from Table 1, unfortunately, we find that the coherence hypothesis of pretrained LMs in prior work does not always hold in practice: it is likely to produce incoherent sentences when powerful LMs rigorously follow model confidence at each step with greedy algorithm.

## 4 Proposed Method: *Look-back*

As presented in Algorithm 1, *Look-back* first leverages probability distribution distance between current and prior steps to avoid repetitions (§4.1), then incorporates reference from given prefix to mitigate topic drifts (§4.2).

## 4.1 Alleviating Repetitions with Reference from Prior Texts

**Signal for Surface or Semantic Repetitions** In the decoding process of open-ended text generation, one of the plausible tokens is selected/sampled according to model probability. Inspired by the decisive role of probability distribution, we investigate measuring the distance between current and prior steps in disbrituion space via KL divergence: $D_{\mathrm{KL}}(p_t|p'_t)$ for any $1 \leq t' < t$. As the distance

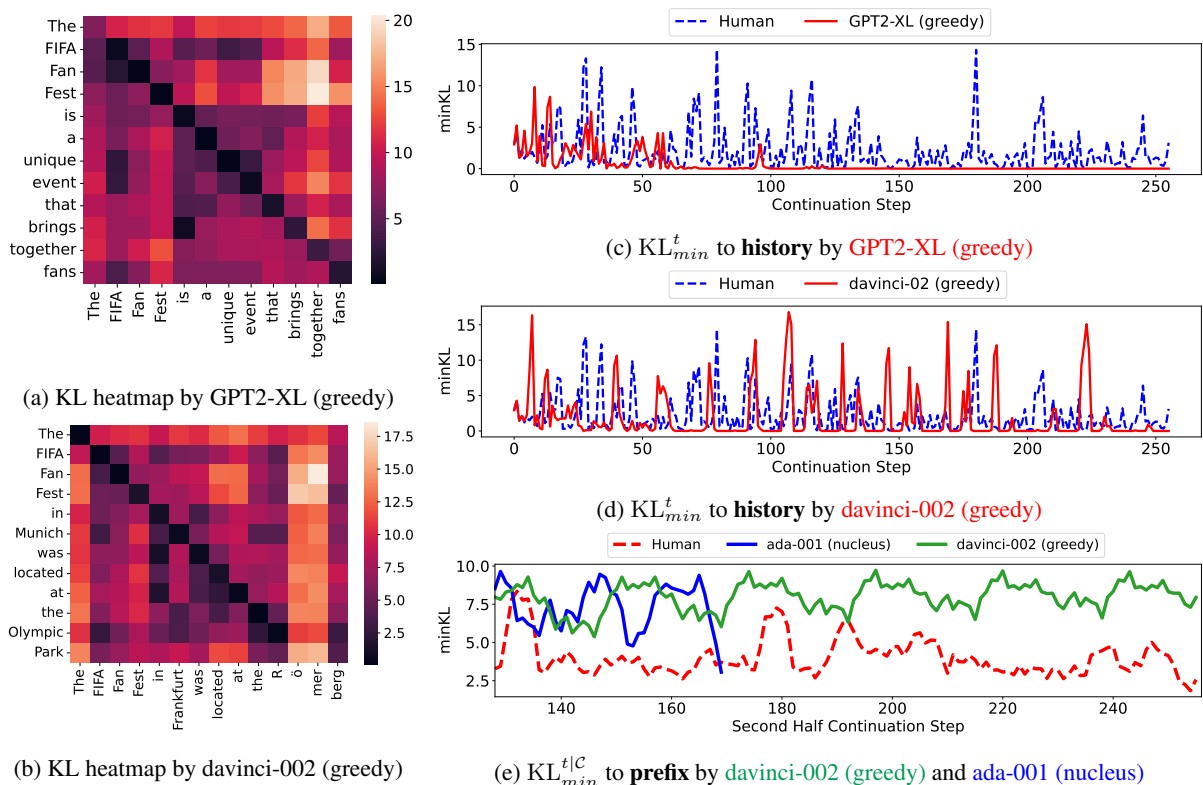

(a) KL heatmap by GPT2-XL (greedy)

(b) KL heatmap by davinci-002 (greedy)

(c) $\text{KL}_{min}^{t}$ to **history** by GPT2-XL (greedy)

(d) $\text{KL}_{min}^{t}$ to **history** by davinci-002 (greedy)

(e) $\text{KL}_{min}^{t|\mathcal{C}}$ to **prefix** by davinci-002 (greedy) and ada-001 (nucleus)

Figure 2: Probability distribution distance of GPT2-XL measured by KL divergence for repetitive (a,b,c,d) and off-topic (e) continuation presented in Table 1. (a) and (b): Dark cells along diagonal indicate that steps of small distance with history tend to produce repetitive tokens. (c) and (d): Compared with human continuation, minimum distribution distance with past gradually approaches 0 (red curves) as similar phrases keep repeating during decoding. (e): distribution of incoherent continuation (green and blue curves) is prone to stay farther from given prefix as decoding proceeds.

---

**Algorithm 1** *Look-back* Decoding

**Input:** Prefix $\mathcal{C} = \{x_1 \ldots x_m\}$, language model with vocabulary $V$, beam size $k$ and threshold $\alpha$
**Output:** Continuation $\mathcal{G} = \{x_{m+1} \ldots x_{m+n}\}$

$\quad \mathcal{G} \leftarrow \{\}$
$\quad$ **for** $m + 1 \leq t \leq m + n$ **do**
$\quad\quad$ **if** $\text{KL}_{min}^{t} \leq \alpha$ **then** $\quad \triangleright$ Alleviate Repetitions
$\quad\quad\quad$ **for** $v \in V^k$ **do**
$\quad\quad\quad\quad q_v = \text{softmax}(-\text{KL}_{min}^{t+1,v|\mathcal{C}})$
$\quad\quad\quad$ **end for**
$\quad\quad\quad x_t = v \sim q_v \quad\quad \triangleright$ Improve Coherence
$\quad\quad$ **else**
$\quad\quad\quad x_t = \text{argmax}_{v \in V} \, p_\theta(v|x_{<t})$
$\quad\quad$ **end if**
$\quad\quad \mathcal{G} \leftarrow \mathcal{G} \cup \{x_t\}$
$\quad$ **end for**

---

heatmap shown in Figure 2a, for steps generating identical tokens, their corresponding probability distributions stay close to each other than those with dissimilar outputs.

Note that neither the contrastive training objec-

tive (SimCTG) (Su et al., 2022) nor its contrastive search decoding algorithm (Su and Xu, 2022) can be directly applied to LLMs such as GPT3, where its hidden states are inaccesible. Fortunately, we can directly detect surface or semantic repetitions from GPT3 by analyzing available probability distribution: step pairs producing either identical token or tokens sharing similar semantic meaning are distinguishable with distribution distance. Take Figure 2b as an instance: output token pairs from decoding steps with closest probability distributions are the 1st and 2nd *FAN*, city *Munich* and *Frankfurt*, location *Olympic* and *R* of *Römerberg*.

As repetitive steps tend to stay extremely close to prior steps with similar outputs in probability distribution space, we calculate the probability distribution distance between the $t$-th and closest prior step as $\text{KL}_{min}^{t}$ for further analysis:

$$\text{KL}_{min}^{t} = \min_{1 \leq j \leq t-1} \text{KL}\left(p(\cdot|x_{<t}) \| p(\cdot|x_{<j})\right)$$

As demonstrated in Figure 2c and Figure 2d, values of $\text{KL}_{min}^{t}$ become flat as repetition-style degenera-

tion advances[4].

**Alleviating Repetitions** Since identical or similar repetition pattern could be forecasted via probablity distribution analysis, *Look-back* attempts to avoid repetitive sentences or phrases prior to actual generation. Practically, when $\text{KL}_{min}^t$ has been below a pre-defined threshold $\alpha$, an alarm is triggered and *Look-back* attempts to sample a token from the top-$k$ most probable tokens from the vocabulary $V$ rather than sticking to the top-1 token:

$$x_t \begin{cases} \sim \text{Unif}(V^k), & \text{if } \text{KL}_{min}^t \leq \alpha \\ = \arg\max_{v \in V} p_\theta(v|x_{<t}), & \text{Otherwise} \end{cases}$$

where $V^k$ is the set of top-$k$ most probable tokens from the vocabulary $V$. To avoid false positive cases where one step identified with high possibility to repeat may not necessarily lead to undesired repetitions, we do not exclude its most probable token from the plausible candidate set on purpose.

### 4.2 Improving Coherence with Reference from Given Prefix

**Signal for Topic Drift** In open-ended generation, in order to produce sentences coherent with the given prefix, the decoding algorithm is required to provide further elaboration of the major topic conveyed in the prefix. According to the prior observations (e.g., *Munich* and *Frankfurt* in Figure 2b), decoding steps with tokens sharing similar semantic meaning are close to each other with respect to probability distribution distance. Therefore, we explore the KL divergence between current and prefix $m$ steps that should keep to the same topic:

$$\text{KL}_{min}^{t|\mathcal{C}} = \min_{1 \leq j \leq m} \text{KL}\left(p(\cdot|x_{<t})\|p(\cdot|x_{<j})\right)$$

When comparing distribution distance of incoherent generation with natural continuation to the same prefix, the probability distribution divergence maintains a much higher level for generation with obvious topic drift, as shown in Figure 2e.

**Improving Coherence** When the model is prone to provide repetitive tokens, one straightforward solution for avoiding repetition is to randomly sample from the top-$k$ plausible tokens. It is likely to result in unnatural topic drift due to undesired sampling choices accumulation over long sequence

---

[4]Spikes in Figure 2d in later decoding steps correspond to multiple tokens for representing one single location, e.g., *ö, mer, berg* for Römerberg in Figure 2b.

decoding, which is frequently observed in sampling algorithms (Eikema and Aziz, 2020; Maynez et al., 2020). On the other side, the probability distribution distance between current and prefix is able to distinguish whether the generation is on-topic or not. Therefore, *Look-back* wisely samples from the plausible candidates according to their influence on coherence reflected by next-step distribution distance with prefix:

$$\text{KL}_{min}^{t+1,v|\mathcal{C}} = \min_{1 \leq j \leq m} \text{KL}\left(p(\cdot|x_{<t+1}, v)\|p(\cdot|x_{<j})\right)$$

$$x_t \begin{cases} \sim \text{softmax}(-\text{KL}_{min}^{t+1,v|\mathcal{C}}), & \text{if } \text{KL}_{min}^t \leq \alpha \\ = \arg\max_{v \in V} p_\theta(v|x_{<t}), & \text{Otherwise} \end{cases}$$

where tokens with larger next-step distance to prefix is less likely to be sampled given the softmax operation upon KL divergence.

## 5 Experiments

In this section, we first introduce the datasets (§5.1) and automatic metrics (§5.2) used to evaluate the generation quality of the proposed *Look-back* and other strong decoding baselines (§5.3). We then analyze experimental results evaluated by automatic metrics (§5.5) and human evaluators (§5.6). Lastly, we show effectiveness of different techniques used in *Look-back* through detailed analyses (§5.7).

### 5.1 Datasets

We consider two applications of open-ended text generation: *1) document continuation* on WikiText-103 with articles fitting the Good or Featured article criteria specified by editors on Wikipedia (Merity et al., 2016), and *2) story generation* on WritingPrompts, which is a challenging task for inspiring continuations with abstract, high-level story prompts submitted by online users and continuations responded by others freely on Reddit (Fan et al., 2018).

### 5.2 Evaluation Metrics

We adopt the following automatic metrics to evaluate generation quality:

**Repetition** We use *rep-n* to measure sequence-level repetition according to the portion of duplicate $n$-grams (Welleck et al., 2019). For a sequence $x$, $rep\text{-}n = 1.0 - \frac{|\text{unique n-grams(x)}|}{|\text{total n-grams(x)}|}$.

**Diversity** Following (Su et al., 2022), we obtain an overall assessment of model repetition by considering repetition at different $n$-gram levels: $diversity = \prod_{n=2}^4 (1.0 - \text{rep-}n)$.

| LM | Decoding | WikiText-103 | | | | | | WritingPrompts | | | | | |
|---|---|---|---|---|---|---|---|---|---|---|---|---|---|
| | | rep-2 ↓ | rep-3 ↓ | rep-4 ↓ | diversity ↑ | MAUVE ↑ | coherence ↑ | rep-2 ↓ | rep-3 ↓ | rep-4 ↓ | diversity ↑ | MAUVE ↑ | coherence ↑ |
| | human | 6.91 | 1.83 | 0.70 | 0.91 | - | 0.62 | 15.61 | 3.78 | 1.24 | 0.80 | - | 0.31 |
| GPT2-XL | nucleus | 5.29 | 1.97 | 1.42 | 0.92 | 0.69 | 0.53 | 5.40 | 2.41 | 1.72 | 0.91 | 0.22 | 0.34 |
| | typical | **3.61** | **1.07** | 0.73 | **0.95** | 0.70 | 0.50 | 3.60 | 1.51 | 1.10 | 0.94 | 0.19 | 0.30 |
| | $\eta$-sampling | 6.25 | 2.49 | 1.80 | 0.90 | 0.68 | 0.55 | 6.17 | 2.88 | 2.16 | 0.89 | 0.17 | 0.35 |
| | SimCTG | 5.37 | 1.97 | 1.46 | 0.91 | 0.72 | 0.53 | **2.84** | **0.36** | **0.19** | **0.97** | 0.18 | 0.31 |
| | *Look-back* | 8.22 | 1.34 | **0.38** | 0.90 | **0.81** | **0.65** | 7.94 | 1.25 | 0.33 | 0.91 | **0.24** | **0.52** |
| OPT-6.7B | nucleus | 6.08 | 2.19 | 1.43 | **0.91** | 0.63 | 0.56 | 5.82 | 3.12 | 2.57 | 0.89 | 0.13 | 0.33 |
| | typical | 6.58 | 2.25 | 1.37 | 0.90 | 0.61 | 0.57 | 5.80 | 2.67 | 1.93 | 0.90 | 0.14 | 0.33 |
| | $\eta$-sampling | 6.07 | 2.26 | 1.55 | 0.90 | 0.66 | 0.56 | **4.72** | **1.93** | 1.36 | **0.92** | 0.15 | 0.34 |
| | SimCTG | **5.44** | 1.97 | 1.38 | **0.91** | 0.56 | 0.55 | 7.49 | 4.25 | 3.10 | 0.86 | 0.08 | 0.20 |
| | *Look-back* | 9.21 | **1.74** | **0.53** | 0.89 | **0.80** | **0.65** | 9.77 | 2.18 | **0.74** | 0.88 | **0.19** | **0.43** |

Table 2: Automatic evaluation results of different decoding algorithms for document continuation and story generation. Continuation generated by *Look-back* is of similar level of diversity as human texts while much more relevant to prefix (highest *coherence*) and semantically similar to human continuation (highest *MAUVE*).

**MAUVE**   By computing information divergences in a quantized embedding space[5], *MAUVE* (Pillutla et al., 2021) directly compares the learnt distribution from a text generation model to the distribution of human-written continuation.

**Coherence**   The semantic *coherence* between prefix and continuation is measured as the cosine similarity between their sentence embeddings represented by SimCSE (Gao et al., 2021).

Results measured by all metrics range from 0 to 1, and higher scores indicate better generation except *rep-n*, for which the lower the better.

### 5.3   Decoding Baselines

Given pretrained LMs with conventional MLE, we evaluate *Look-back* together with various decoding algorithms for fair comparisons.

**Search Methods**   We consider the competitive contrastive search proposed in *SimCTG* (Su et al., 2022) that predicts the next token based on both the output distribution and representation similarities between candidates and past tokens[6].

**Sampling Methods**   *Nucleus* sampling (Holtzman et al., 2019) samples the next token from the top-$p$ portion of the probability mass. *Typical* decoding (Meister et al., 2022) samples from the set of words whose negative log-probabilities are close to the conditional entropy. *$\eta$-sampling* (Hewitt et al., 2022) truncates any word whose probability is smaller than an entropy-based threshold.

### 5.4   Implementation Details

We randomly sample 1,000 instances from the original training data of WikiText-103 and Writing-Prompts as our validation and test sets. Given the beginning several tokens as prefix[7], we generate 256 tokens with different decoding algorithms and disregard those after the end-of-text token during evaluation. Practically, we consider a sliding window comprising 128 prior tokens to avoid undesired repetitions while allow necessary repetitions of text far from the current decoding step. We perform experiments with pre-trained LMs from different families and scales: GPT2-XL (Radford et al., 2019) and OPT-6.7B (Zhang et al., 2022). The same set of hyperparameters is used to decode from different LMs: the beam size for *beam* search is 10, $p = 0.95$ for *nucleus*, $\tau = 0.92$ for typical, and $\eta = 0.0003$ for *$\eta$-sampling*. We follow the recommended range for $k = \{5, 8, 10\}$ and $\alpha = [0.5, 0.9]$ in *SimCTG* and select the set based on their MAUVE scores on the validation set. For *Look-back*, the range of candidate amount $k$ is $\{5, 8, 10\}$ and the threshold $\alpha$ is ranging from $[0.5, 1.6]$. We select hyperparameters that result in the *rep-2* score closest to human's and the optimal MAUVE performance on the validation set.

### 5.5   Results

In Table 2, we show the performance of different decoding algorithms as well as natural human continuation evaluated by automatic metrics. On both datasets, *Look-back* consistently achieves the highest *MAUVE* scores and *coherence* scores, which indicates that the generation of *Look-back* has token

---

[5]We use GPT2-XL for text sequence embedding.

[6]We disregard *greedy* and *beam* search as they kept producing repetitive phrases/sentences in prior studies (Welleck et al., 2019; Holtzman et al., 2019).

[7]First 32 tokens are used as prefix for WikiText-103, while the original prompts are used for WritingPrompts.

| LM | Criterion | *Look-back* better (p-value) | same | SimCTG better |
|---|---|---|---|---|
| GPT2-XL | Fluency | 0.46 (.0127) | 0.27 | 0.27 |
| | Coherence | 0.57 (.0004) | 0.16 | 0.27 |
| OPT-6.7B | Fluency | 0.38 (.0508) | 0.37 | 0.25 |
| | Coherence | 0.53 (.0078) | 0.16 | 0.31 |

Table 3: Human Evaluation on generations from *Look-back* and the second best SimCTG with examples sampled from WikiText-103. Continuation generated by Look-back is preferred to SimCTG significantly by human evaluators w.r.t both fluency and coherence.

distribution closeness with human continuations while staying relevant to the given prefixes. Meanwhile, *Look-back* is capable of producing texts with similar repetition and diversity level as the natural human text, which implies the fluency and informativeness of the generated text. We also notice that generations from all decoding algorithms obtain relatively low *MAUVE* and *coherence* scores on WritingPrompts. This is because the given prefixes are abstract and the human written references are diverse and varied, which results in low *coherence* and *MAUVE* w.r.t. various model continuations.

## 5.6 Human Evaluation

To further evaluate the quality of generated texts, we randomly sample two sets of 50 examples from WikiText-103 to produce prefixes for GPT2-XL and OPT-6.7B respectively and generate continuations from them. Then, we ask 3 evaluators to compare generated continuations from *Look-back* and the second best baseline SimCTG in two dimensions: *1) fluency*: diverse and natural content without repeated words, phrases or sentences; *2) coherence*: well-organized and easy to follow; being consistent with the topics presented in the human-written prefix without abrupt topic drifts. We ask annotators to choose one out of three options: the 1st continuation is better, the 2nd is better, or the two are of the same quality. As presented in Table 3, for both evaluation dimensions, the content generated by *Look-back* is preferred or marked as equally good by evaluators around or more than 70% of the time compared with baseline, which aligns well with the automatic metrics in Table 2.

## 5.7 Further Analyses

In this section, we analyze the effectiveness of different techniques used by *Look-back* individually.

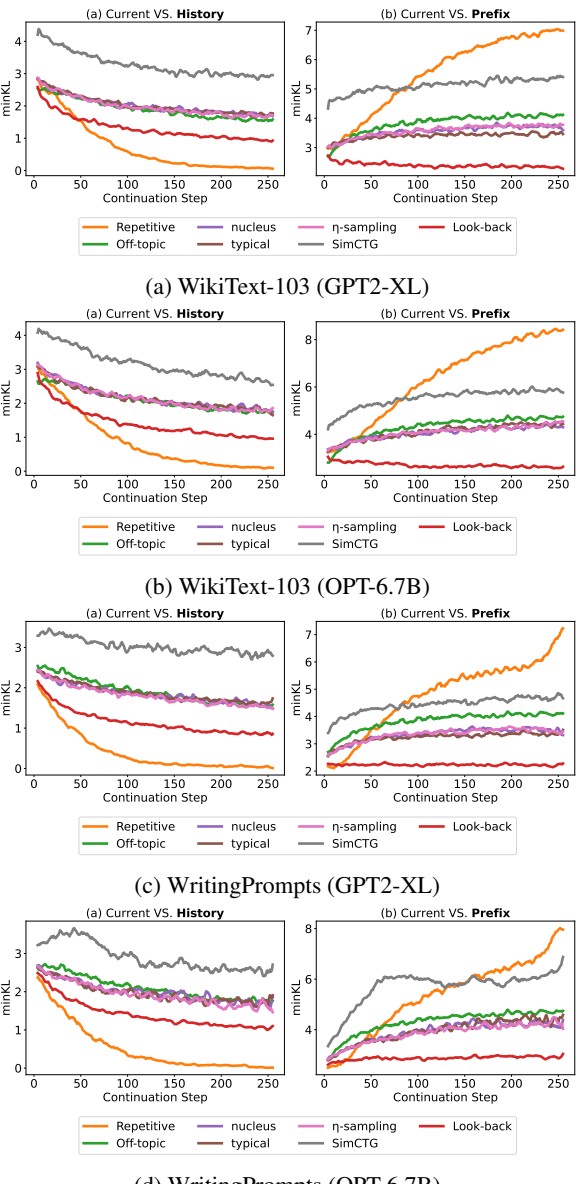

(a) WikiText-103 (GPT2-XL)

(b) WikiText-103 (OPT-6.7B)

(c) WritingPrompts (GPT2-XL)

(d) WritingPrompts (OPT-6.7B)

Figure 3: Minimum KL divergence between current step and (a) **history** or (b) **prefix** from GPT2-XL and OPT-6.7B decoded by different algorithms on the test set of WikiText103 and WritingPrompts. Probability distribution of *Look-back* keeps distance to history to avoid repetitions but stays close to prefix to guarantee coherence.

**Analyzing Probability Distribution Distance.** To verify whether decoding with *Look-back* appropriately constrains the probability distribution distance to past steps, we compare $\text{KL}_{min}^{t}$ to history and $\text{KL}_{min}^{t|\mathcal{C}}$ to prefix of degeneration and different decoding algorithms in Figure 3. Although all improved decoding algorithms keep distance to historical probability distribution to avoid repetitions compared with greedy algorithm (Repetitive in the left column of Figure 3, the probability dis-

| LM | Sampling | diversity | MAUVE | coherence |
|---|---|---|---|---|
| | | WikiText-103 | | |
| GPT2-XL | Uniform | **0.93** | 0.71 | 0.61 |
| | Softmax | 0.90 | **0.81** | **0.65** |
| OPT-6.7B | Uniform | **0.93** | 0.60 | 0.52 |
| | Softmax | 0.89 | **0.80** | **0.65** |
| | | WritingPrompts | | |
| GPT2-XL | Uniform | **0.93** | 0.15 | 0.45 |
| | Softmax | 0.91 | **0.24** | **0.52** |
| OPT-6.7B | Uniform | **0.91** | 0.14 | 0.29 |
| | Softmax | 0.88 | **0.19** | **0.43** |

Table 4: Effects of probability distribution-guided sampling of *Look-back* (Softmax) on generation quality. With similar level of diverse content as human text, *Look-back* samples according to softmax of negative distribution distance to prefix, leading to improved coherence compared with Uniform.

tribution of *Look-back* (*Look-back* in the right column of Figure 3 is much closer to the given prefix, which distinguishes it from off-topic continuation compared with other algorithms.

**Softmax vs. Uniform.** According to the softmax operation on $\mathrm{KL}_{min}^{t|\mathcal{C}}$ introduced in §4.2, the closer the next step's probability distribution to prefix, the more likely the corresponding plausible token is selected to avoid undesired topic drift compared with random sampling. In Table 4, we empirically investigate the impact of plausible token sampling, uniform vs. softmax, on generation quality and find *Look-back* significantly enhances coherence on both datasets compared with random sampling. Although diversity drops with distribution distance-guided sampling in *Look-back*, both sampling strategies produce similar level of diverse content as human texts listed in Table 2.

**Effects of Candidate Amount and Threshold $\alpha$.** In §4.1, the hyperparameter $\alpha$ determines whether the current step is likely to produce repetitive continuation while $k$ restricts the range of plausible token candidates. The second best baseline Sim-CTG has the similar candidate amount parameter $k$ and the $\alpha$ to balance model confidence and degeneration penalty. When GPT2-XL is used to decode with *Look-back* and SimCTG on WikiText-103, we visualize the impact of hyperparameters on generation quality in Figure 4 and Figure 5. The $\alpha$ in *Look-back* is different from that in SimCTG, but both control reliance on model confidence: a larger

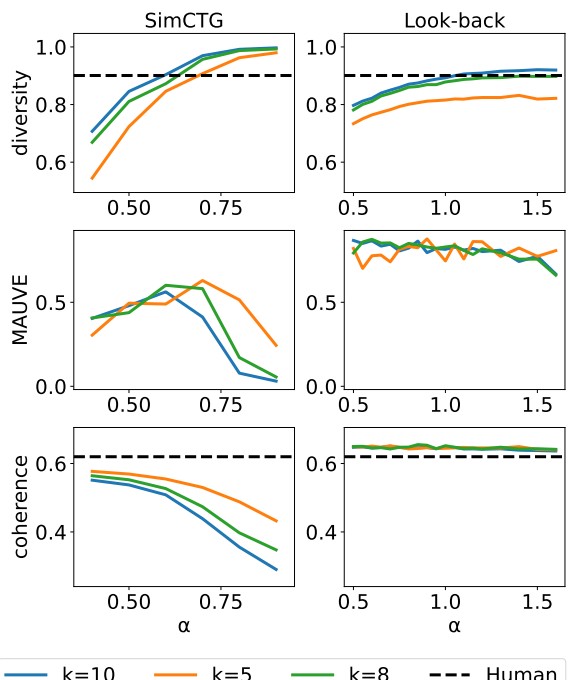

Figure 4: Impact of decoding hyperparameters on validation set of WikiText-103. Compared with the other search algorithm SimCTG (1st column), *Look-back* (2nd column) keeps relatively higher MAUVE and coherence scores regardless of plausible token amount $k$ and the $\mathrm{KL}_{min}^{t}$ threshold $\alpha$. See Figure 5 for more results in other settings.

$\alpha$ indicates the most probable token is less likely to be adopted, hence more diversity is obtained. We also observe that for *Look-back*, the relevance of generated text to prefix (high *coherence*) and human continuation (high *MAUVE*) is much more robust to various hyperparameter values compared with SimCTG.

## 5.8 Case Study

Given a prefix sampled from WikiText-103, we present truncated human continuations as well as generations from *Look-back* and SimCTG in Table 5 and leave more examples in Appendix Table 6. The prefix is talking about the design of a race car game. Both human and *Look-back* continuations focus on describing major difficulties encountered during the game design, while SimCTG switches to a different topic by pointing to an online introduction of the game in the second half of continuation. Interestingly, *Look-back* explains how the team of more than twenty people was formed, which is coherent with the topic in the prefix.

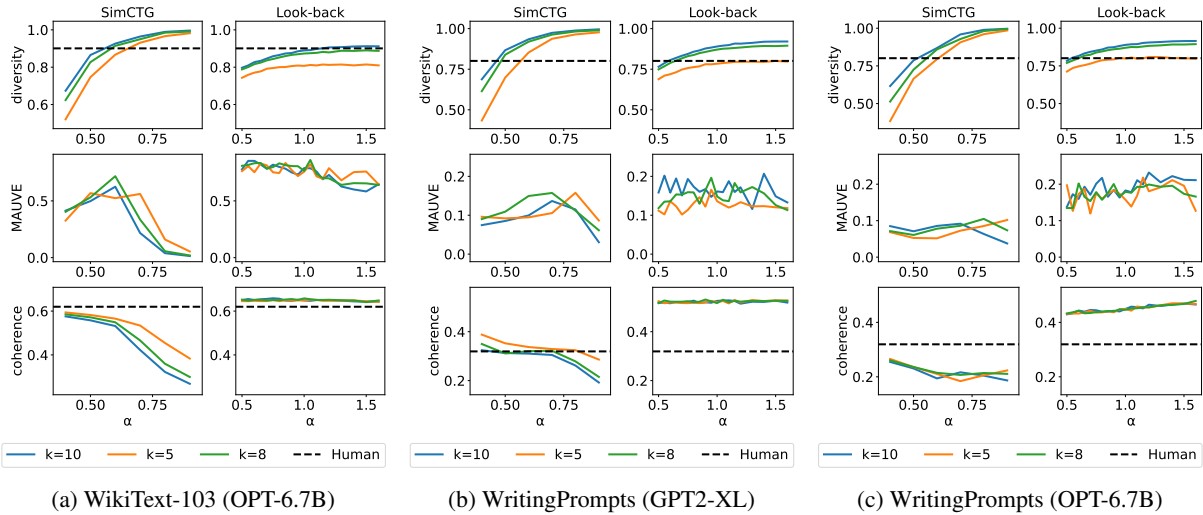

(a) WikiText-103 (OPT-6.7B)  (b) WritingPrompts (GPT2-XL)  (c) WritingPrompts (OPT-6.7B)

Figure 5: (Continuation from Figure 4) Impact of decoding hyperparameters on validation set of WikiText103 and WritingPrompts.

---

**Prefix**: Ridge Racer Revolution was developed over eight months by a team of more than twenty people, most of whom joined just for Ridge Racer Revolution. The biggest difficulties were

**Human**: the link-up mode, rear-view mirror, and running the game at high speeds. Each member had worked on other console games, and they commented that the graphics detail showed the improvements in skill and technique. The team wanted players to enjoy a more in-depth game than the original, so rather than develop a port of Ridge Racer 2, as many features as possible were added. The early designs for the new courses were made...

**SimCTG**: balancing the game, which had to be a mix of arcade and racing elements, along wtih making an open world that felt like the future of video games.\n\nIn order to do this, we used Unreal Engine 3, the game engine that powers ... *You can learn more about the game by clicking here, but be warned, there are spoilers in this article. If you're planning on reading this article, I suggest you stop reading now before it spoils the game for you...*

**Look-back**: the lack thereof: the original game was built in a single year; Ridge Crash took more. The original developers were all gone, and the original team of programmers, artists, and designers from Ridge Revolution, including the lead programmer at Capcom, had all left the company by 2007...In the end, a new team of twenty-five employees was assembled for Revolution, who took nearly two years to complete the game. In all, this team of more...

Table 5: Case study of an instance sampled from WikiText-103 with GPT2-XL. Continuation of both human and *Look-back* discusses difficulties in game design, while SimCTG gradually produces less informative sentences with slight topic drift to game introduction (in pink ). Refer to Table 6 for more examples.

## 6 Conclusion

The distance between output distributions signals potential failure modes of text generation, such as dull repetition and incoherence. We propose *Look-back*, a novel decoding algorithm that utilizes the KL divergence between the current and historic decoding steps, to steer the output distribution into a plausible subspace. *Look-back* can generate higher-quality text and outperforms several strong decoding algorithms in both automatic and human evaluation. However, KL divergence may not be the optimal measure for text output distributions and we leave the investigation of other measures to future work. In addition, the idea proposed in this work can also be used for other specialized constrained decoding scenarios, such as preventing hallucination.

## Limitations

We discuss the limitations of our work as follows:

- *Look-back* penalizes next tokens that result in low KL divergence with historic output distributions. However, we can not explicitly distinguish if such tokens are natural or undesired repetitions. This may lead to aggressive eliminations of possible outputs. We leave the distinction of different repetitions to future work.

- *Look-back* tends to show a higher bi-gram repetition score than other decoding methods because it encourages the coherence with prefix text at each decoding step. As we use a short prefix text following previous evaluation protocol, which might not be sufficiently informative, we will adopt a more comprehensive evaluation setup in the future or prepend relevant text in the beginning at decoding time.

- Most of our evaluations rely on automatic metrics, such as MAUVE scores. However, we found that these metrics may not truthfully reflect the quality of text, for example, MAUVE score is sensitive to the choice of sentence embedding models. In general, open-ended text generation still poses a great challenge to the development of NLG algorithms.

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

| Dataset (LM) | Prefix/Continuation |
|---|---|
| WikiText-103 (OPT-6.7B) | **Prefix**: A new show in London in 1912 showcased the best of music hall\'s talent. The Royal Command Performance took place at the Palace Theatre in London, which was |
| | **Human**: managed by Alfred Butt. The show was organised by Oswald Stoll, an Australian impresario who managed a string of West End and provincial theatres. Stoll, although a fan of Lloyd's, disliked the vulgarity of her act and championed a return to a more family @-@ friendly atmosphere within the music hall. Because of this, and her participation in the earlier music hall war, Stoll left her out of the line @-@ up. He placed an advert in The Era on the day of the performance warning that " Coarseness and vulgarity etc are not allowed... |
| | **SimCTG**: one of the most popular theatres in the West End at the time. *Start the conversation, or Read more at BroadwayWorld.com.I have been waiting for this. Thank you for putting it together. You should cross post to /r/blunderyears as well...* |
| | **Look-back**: the home of the Royal Commandos during the First World War. The show starred the likes of Harry Lauder, who played the role he was born to play, 'The King in Yellow', and Fred Karno - who, as the 'King', was the star attraction. It was a huge success, and the Royal Variety Performance took its spot in the calendar. It's a tradition that continues to this day - and the King in Yellow is still a big draw at any show... |
| WritingPrompts (GPT2-XL) | **Prefix**: The world is teetering on the brink of destruction . Only 2 people remain alive , both of them old ladies . They are face to face with the eldritch creature that has singlehandedly destroyed their planet . And in its terrifying presence ... they make cheeky small talk about what brought them here . |
| | **Human**: FADE IN : EXT . A DESOLATE LANDSCAPE - AFTERNOON *A red sky covered with black clouds hangs above an expanse of blasted countryside . Dark craters pockmark the land , and the corrupted hulks of destroyed war machines lie rusting in the acrid air . Amidst this desolation , two elderly women sit on rocking chairs that are curiously undamaged . These are DELILAH and TABITHA . A small table with a tea set is present between them . * **DELILAH : ** It 's a bit hot today , is n't it ?... |
| | **SimCTG**: *<\|endoftext\|>A group of high school students in New York's Nassau-Suffolkshire school district are demanding to be excused from participating in "Star Wars," "Harry Potter" and Disney movies because it's the holiday season...* |
| | **Look-back**: \n\nThe two old ladies, who were not named in either the book or film, had been visiting from the future when their ship was destroyed. They are not aware of this, but one has become a time traveler from Earth and knows that they were never meant be together in life but are now destined by fate together. One day they will meet again at last on some other planet where one of each woman will be born with powers over time which will allow them each one chance to save the world, which is now destined to fall into the hands of their enemy. One day both are going in the same direction. And one woman has decided that she needs to see to it that they both get there first!... |
| WritingPrompts (OPT-6.7B) | **Prefix**: For the first time ever , a person is born with a genuine superpower . They proceed to live out their entire life without noticing or realizing it . |
| | **Human**: For Francis , the good had always gone hand-in-hand with the bad . He had spent his life trying his best to be a good person , or at least what he believed would be a good person . He realized now that he was n't really sure anymore . When his mother left his dad , Francis had chosen to stay with him . His friends had all told him he was smart to stay with the rich parent , but that had n't been the reason , had it ? At the time , he had told himself that his father needed him more . He did . Dad had been really bad at taking care of himself . Without his mother , someone had to be there to deal with dad when he was lonely and angry and ... |
| | **SimCTG**: What if they have superpowers in real life?\nI'm going to need proof of this before I believe it. You know, for science. It's true. I saw it on the internet. So it must be true. *How to Get Rid of Pimples on Forehead...* |
| | **Look-back**: The person is born with the ability not necessarily in the form of super strength or super speed (although they may have those abilities as well) but the superpower is the ability to see the future! They see the outcomes in their life and they live their days according to the outcome they see for themselves! In other words, they plan their life out according how their futures will unfold according. They see what will happen and they do what they can so that it will come. And because they live their whole live without realizing they have this ability they are able live life to the fullest. Because the future is already set for them, their entire life unfolds according. Their whole life, from the moment of their birth, unfolds according as planned in their mind... |

Table 6: Case study of instances sampled from WikiText-103 and WritingPrompts. Unnatural topic drifts are frequently observed in generations from SimCTG (in pink).