# OpenReview forum: "Look-back Decoding for Open-Ended Text Generation"
_EMNLP/2023/Conference — EMNLP 2023 Main_

### Official Review · Reviewer_7Euy · 2023-07-21

**Soundness:** 4

**Excitement:**

4: Strong: This paper deepens the understanding of some phenomenon or lowers the barriers to an existing research direction.

**Missing References:**

- Recent work _The Stable Entropy Hypothesis and Entropy-Aware Decoding: An Analysis and Algorithm for Robust Natural Language Generation_ (https://arxiv.org/abs/2302.06784v1) also aims to propose a novel decoding algorithm which mimics human probability distribution.

**Paper Topic And Main Contributions:**

The work describes a new and simple decoding algorithm, which pays special attention to the KL divergence between the current vocabulary distribution and past ones. This method is very well justified and described formally as well as evaluated on text generation.

**Reasons To Accept:**

The paper proposes a very simple decoding mechanism which has clear and intuitive interpretation (distribution distance), is motivated by linguistics and at the same time performs very well for unconstrained text generation, as evaluated by the authors. The paper is well-structured and written and could be accepted as-is.

**Reasons To Reject:**

While there is some overlap with existing work (see missing references) conceptually, the papers are different enough to both warrant a place.

**Reproducibility:**

4: Could mostly reproduce the results, but there may be some variation because of sample variance or minor variations in their interpretation of the protocol or method.

**Reviewer Confidence:**

2: Willing to defend my evaluation, but it is fairly likely that I missed some details, didn't understand some central points, or can't be sure about the novelty of the work.

**Typos Grammar Style And Presentation Improvements:**

- Abstract: _don't_ $\rightarrow$ _do not_.

---

> ### Author Rebuttal · Authors · 2023-08-29
>
> **Major difference with entropy-aware decoding**
>
> Both works aim to address degeneration issues in open-ended text generation, but distinct decoding strategies are proposed according to different signals that indicate good generation or unwanted degeneration. 1) **entropy-aware decoding** observes the strong correlation between good generation quality and narrow **entropy** zone, hence entropy-aware decoding is proposed to promote good generation by constraining greedy decoding into the narrow entropy zone; 2) **Look-back** finds that smaller **KL divergence** between current and prior steps is potential signal for repetitions and larger KL divergence between current and prefix step indicates potential topic drift. Therefore, we restrict the next token probability distribution within a plausible distance to the history to avoid repetitions and topic drifts. We will add prior discussion in our related work.

---

### Official Review · Reviewer_Ch7J · 2023-08-03

**Soundness:** 4

**Excitement:**

4: Strong: This paper deepens the understanding of some phenomenon or lowers the barriers to an existing research direction.

**Paper Topic And Main Contributions:**

The paper proposes a solution to reduce repetitiveness in open-ended text generation, but at the same time to make the generated text more coherent with the given prefix, avoiding topic drifts.

The approach is based on the KL-divergence of the probability distribution for the next word to be predicted and the probability distribution of already predicted words.

The paper provides experiments on two tasks and compares with other methods which go beyond greedy search and beam search to avoid repetitions.



**Questions For The Authors:**

When computing the (minimum) KL-divergence, you always look back until the first token. Would it make sense to have some kind of a sliding window of N previously generated words so that e.g. the generation remains coherent not only given the prefix, but also what has been generated after it? The same for repetitions - maybe you want to allow some repetitions at a larger distance in the text?



**Reasons To Accept:**

It is a well-written paper with a good and novel (at least for me who is not an expert in the field) approach. The approach is evaluated well, including human evaluation. There are also quite substantial additional experiments which show the choice of the hyperparameters (thresholds for KL-divergence). The heatmaps for KL give a good motivation for the approach. The approach seems to be more  coherent than related work, while at the same time having similar rate of repetitions.



**Reasons To Reject:**

In the end, the approach is a heuristic applied at inference time only; a better approach would probably be to modify the attention mechanisms of the underlying model (and the training criterion) so that the model learns to keep coherence with the prefix and avoid repetitions by attending to the previously generated words (and the training would probably have to have some way of mitigating exposure bias so that such unwanted repetitions would actually be observed). As I am not an expert in the field, I am not sure whether such approaches exist; but if the goal was just to modify the search at inference time, then the authors did a good job about it.

And I have doubts about the example presented in Table 5: clearly, the proposed look-back method does not make a natural continuation of the prompt: "...the biggest difficulties were..." is continued with "the lack thereof: ". Are there maybe any better examples? Maybe the authors should elaborate whether their method in most cases starts with a natural continuation or not.


**Reproducibility:**

3: Could reproduce the results with some difficulty. The settings of parameters are underspecified or subjectively determined; the training/evaluation data are not widely available.

**Reviewer Confidence:**

3: Pretty sure, but there's a chance I missed something. Although I have a good feel for this area in general, I did not carefully check the paper's details, e.g., the math, experimental design, or novelty.

**Typos Grammar Style And Presentation Improvements:**

Footnote 1: "copying" is not described clearly, maybe you can re-word this.
Line 244: "alerm" -> "alarm"
Line 283: "between current and prefix": "current word"?
Line 290: maybe you could better explain how the softmax is computed over the KL_min values (the formulas are hard to read, so maybe just explain in words what is going on there).
Line 372: "is [0.5, 1.6]" -> "is ranging from [0.5, 1.6]"
Line 461: "both controls reliance" -> "both control reliance" ?

---

> ### Author Rebuttal · Authors · 2023-08-29
>
> We thank your helpful suggestions and would like to clarify following details:
>
> **Improved learning vs. improved decoding to alleviate degeneration**
>
> As discussed from lines 40 to 50 in Introduction Section and lines 87 to 137 in Related Work Section, there are two types of work to improve generation: 1) **improved learning** that proposes new learning objectives and 2) **improved decoding** that remedies repetitions in searching or combats incoherence in sampling. As pointed from lines 138 to 142, Look-back improves decoding with reference from the given prefix and prior generation without extra effort on fine-tuning LMs. Compared with improved learning methods, Look-back is efficient and feasible to apply to large language models if we do not have access to training or the training is prohibitively expensive. Authors from baseline SimCTG also found that skipping training and directly applying decoding-time objective could work similarly [1].
>
> [1] Su, Yixuan, and Nigel Collier. "Contrastive search is what you need for neural text generation." arXiv preprint arXiv:2210.14140 (2022)
>
> **More case studies of generation**
>
> We provide more examples covering more models and datasets in Table 6 of Appendix, and will release complete generation from all models soon. Continuation from Look-back should be more natural than other baselines: 1) In Table 2, highest **Mauve** scores imply that the text distribution of Look-back is very similar to that of human-written continuation, and 2) higher **fluency** values from human evaluation in Table 3 show generation from Look-back is more fluent than SimCTG.
>
> **Sliding window for coherence and repetitions**
>
> **Coherence**: we are aimed at improving coherence by avoiding topic drift. Hence Look-back refers to prefix tokens so that all decoding steps talk about the same topic as prefix. Therefore, we believe checking prefix tokens alone is enough to keep the continuation on-topic at different decoding steps. To generate 256 tokens, we actually use the sliding window 128 to avoid undesired **repetitions** but allow necessary repetitions of text far from the current decoding step. We will add prior consideration to our implementation section.
>
> **Softmax computation in line 290**
>
> In line 289, we first measure next-step distribution distance with prefix, if one of the top-k plausible tokens, v, is selected at the current step.  Given the softmax over the negative distribution distance in line 290, tokens resulting in larger next-step distance to prefix are less likely to be sampled, hence the generation is more likely to be coherent with prefix.

---

### Official Review · Reviewer_q9mD · 2023-08-05

**Soundness:** 4

**Excitement:**

4: Strong: This paper deepens the understanding of some phenomenon or lowers the barriers to an existing research direction.

**Paper Topic And Main Contributions:**

This paper presents the new decoding algorithms that utilizes KL divergence for coherent and non-repetitive text generation.

**Reasons To Accept:**

1. They utilize KL divergence, guided by the observation of KL's tendency towards semantic similarity.
2. They have good comparison to the most related work (simCTG) and also included human evaluation (not only automatic metrics)
3. The paper is also supported by qualitative results for the coherency test.

**Reasons To Reject:**

1. lack of scalability analysis (for larger models ~ 10B or 100B)
2. KL divergence itself could not be the good measure for semantic similarity check as they mentioned in the conclusion
3. statistical validation (such as t-test, p-value,) is not present.
4. Quite often, the results of Look-back don't seem to be optimal in Table 2. But the coherence is always the best so it wouldn't be a big issue.

**Reproducibility:**

4: Could mostly reproduce the results, but there may be some variation because of sample variance or minor variations in their interpretation of the protocol or method.

**Reviewer Confidence:**

3: Pretty sure, but there's a chance I missed something. Although I have a good feel for this area in general, I did not carefully check the paper's details, e.g., the math, experimental design, or novelty.

**Typos Grammar Style And Presentation Improvements:**

alerm -> alarm
p_t' or p'_t? <- confusing at the section 4.1

---

> ### Author Rebuttal · Authors · 2023-08-29
>
> We sincerely appreciate your constructive feedback and would like to clarify some concerns as follows:
>
> **Scalability Analysis (e.g., decode LLMs with parameters from 10B to 100B)**
>
> At each decoding step, to avoid repetitions in line 248, the KL divergence computed in line 235 measures the distance between the probability distribution of next token and that of historical tokens. Hence the complexity is irrelevant to language model scale, but relevant to history/prompt length for repetition and coherence consideration, as well as the vocabulary size. Basically, the complexity is O(L^2V), where L is sequence length and V is the vocabulary size.
>
> **Other possible measures for semantic similarity check besides KL divergence**
>
> As discussed in Section 4, we are motivated by the observation that KL divergence could be signals for repetitions or topic drifts, hence leveraging KL divergence to improve fluency and coherence. Empirically, we prove that KL divergence leads to better generation than cosine similarity between hidden states used by the baseline SimCTG. Other distance measures, such as Rényi divergence, Jensen–Shannon divergence, could benefit decoding similarly, which we leave for future direction.
>
> **Statistical significance of human evaluation**
>
> From lines 297 to 416, we demonstrate that the content generated by Look-back is preferred to SimCTG by human evaluators w.r.t both fluency and coherence. We add the p-value in the t-test as follows:
>
> | LM | Criterion | Look-back better (p-value) | same | SimCTG better |
> | -------- | ------- | ------- | ------- | ------- |
> | GPT2-XL | Fluency | 0.46* (0.0127) | 0.27 | 0.27 |
> |                 | Coherence | 0.57* (0.0004) | 0.16 | 0.27 |
> | OPT-6.7B | Fluency | 0.38* (0.0508) | 0.37 | 0.25 |
> |	       | Coherence | 0.53* (0.0078) | 0.16 | 0.31 |
>
> *results from Look-back significantly outperform SimCTG across different models and metrics.
>
> **Performance of Look-back w.r.t. repetition and coherence in Table 2**
>
> The repetition values (rep-2, rep-3, rep-4)  are NOT always the lower, the better: avoiding undesired repetitions leads to smaller repetition values, but necessary multi-occurrences of important concepts increase the repetition values. Instead, low repetition values close to human continuation indicate a good trade-off between avoiding redundant content and emphasizing key components. Look-back obtains a similar level of repetition and diversity as human continuation, as well as best Mauve and Coherence performance over baselines.

---

### Meta-Review · Area_Chair_KoYQ · 2023-09-17

**Recommendation:** 5

**Metareview:**

The paper proposes 'Look-Back', a decoding algorithm that leverages KL divergence to track the distribution between current and historical decoding steps in order to manage repetitions and topic drift. The paper is well written, clearly motivating the problem and explaining the proposed decoding algorithm, and demonstrating the performance of the new decoder via methodical evaluations against SimCTG. The experiments show through automatic and human evaluations the validity and advantage of the proposed Look-back decoding algorithm.

---

### Decision · Program_Chairs · 2023-10-07

**Decision:**

Accept-Main

**Comment:**

The paper proposes 'Look-Back', a decoding algorithm that leverages KL divergence to track the distribution between current and historical decoding steps in order to manage repetitions and topic drift. The paper is well written, clearly motivating the problem and explaining the proposed decoding algorithm, and demonstrating the performance of the new decoder via methodical evaluations against SimCTG. The experiments show through automatic and human evaluations the validity and advantage of the proposed Look-back decoding algorithm.